# Integrating AI in Pakistani ESL classrooms: Teachers' practices, perspectives, and impact on student performance

Tahir Saleem[1]*, Aisha Saleem[2], Dr Muhammad Aslam[3]

1 Professor, Department of English, University of Central Punjab, Lahore, Pakistan, 2 Assistant Professor, Department of English, University of Central Punjab, Lahore, Pakistan, 3 Associate Professor, Department of English, University of Central Punjab, Lahore, Pakistan

* tahir.saleem@ucp.edu.pk

## Abstract

The global rise of Artificial Intelligence (AI) in English as a Second Language (ESL) education has shown promise, yet its application in resource-constrained contexts like Pakistan remains underexplored. This study examines the integration of AI tools in Pakistani ESL classrooms, with a focus on (1) teachers' instructional practices, (2) student learning outcomes, and (3) implementation challenges. Using a mixed-methods approach, data were collected through classroom observations, focus group discussions, and pre- and post-tests on vocabulary and writing skills administered to 100 undergraduate students (50 in the experimental group and 50 in the control group) over 16 weeks. The experimental group, taught with AI tools such as Grammarly and QuillBot, demonstrated significantly greater gains in vocabulary (+45%, $d = 1.12$) and writing performance (+46%, $d = 1.03$) compared to the control group. Qualitative findings revealed that while AI tools supported grammar correction and vocabulary enhancement, their effectiveness was limited by infrastructural constraints, insufficient teacher training, and cultural misalignment in language feedback. The study concludes that AI can meaningfully enhance ESL instruction when paired with teacher facilitation and localized design. It offers novel insights into culturally responsive AI integration in under-resourced educational contexts.

## Introduction

The integration of Artificial Intelligence (AI) in English as a Second Language (ESL) education has garnered significant global attention, particularly for its potential to personalize learning, provide immediate feedback, and facilitate differentiated instruction through intelligent systems [1,2]. In developed educational contexts, AI-powered platforms such as Grammarly, QuillBot, and Duolingo have been shown to improve learners' writing fluency, vocabulary acquisition, and learner engagement. For instance, a recent meta-analysis [3] found that AI-based feedback systems led to a 27% increase

**Data availability statement:** Relevant data is provided in the compressed supporting material as a zip file.

**Funding:** The author(s) received no specific funding for this work.

**Competing interests:** The authors have declared that no competing interests exist.

in writing proficiency scores across diverse ESL populations. However, despite these promising outcomes, the application of such technologies in resource-constrained environments like Pakistan remains underexplored.

In Pakistan, ESL classrooms are shaped by sociocultural complexities, infrastructural disparities, and a pedagogical tradition that emphasizes rote memorization over communicative competence [4]. While urban institutions have begun integrating digital technologies, rural schools still face significant barriers such as poor internet connectivity, lack of teacher training, and limited access to AI platforms. Moreover, even in better-equipped institutions, the integration of AI is often hindered by teachers' limited technological proficiency and the absence of localized content tailored to Pakistani learners. Given these contextual realities, it is crucial to explore the meaningful integration of AI into Pakistan's ESL education landscape.

Previous studies have explored the potential of AI in enhancing language skills [5,6], but these investigations have predominantly focused on technologically advanced contexts. Existing research often emphasizes the technical affordances of AI tools without examining how these technologies interact with pedagogical constraints and sociocultural factors specific to under-resourced contexts. Notably absent from the literature is a detailed understanding of how ESL teachers in Pakistan perceive, adapt, and implement AI tools in their classrooms. Moreover, limited empirical studies address how such integration affects specific learner outcomes, such as vocabulary acquisition and writing proficiency, in a localized context. The primary purpose of this study is to investigate how ESL teachers in Pakistan integrate AI tools into their instructional practices, assess the influence of AI integration on students' vocabulary acquisition and writing proficiency, and explore the challenges teachers encounter along with practical solutions to maximize AI's potential in resource-constrained ESL classrooms.

Anchored in constructivist, sociocultural, and adaptive learning theories, the research investigates both the pedagogical affordances and contextual limitations of AI integration. Unlike prior studies [7,8] that either celebrate AI's potential or list its challenges in abstract terms, this study offers grounded, empirical insights based on classroom observations, focus group discussions, and quantitative assessments. It thereby contributes to both theory and practice by contextualizing AI's role in ESL education within a developing country framework. Recent empirical studies emphasize the relevance of this investigation. For instance, Tahir et al. [9] conducted a qualitative study exploring ESL instructors' experiences with AI-powered tools in Pakistani higher education institutions. Their findings revealed that while instructors recognize the potential of AI to enhance student engagement and provide efficient feedback, they also face significant challenges, including limited access to technology and inadequate training. Similarly, Vera [10] examined the integration of AI tools in university-level ESL teaching, highlighting benefits such as personalized instruction and increased student engagement, alongside challenges like technical issues and over-reliance on technology. These studies illuminate the complex landscape of AI integration in Pakistan's ESL education, emphasizing the need for context-sensitive strategies that address opportunities and obstacles.

The urgency of this research is emphasized by the increasing digitalization of education post-COVID-19 and the national push by Pakistan's Higher Education Commission to adopt emerging technologies. Given the rapid adoption of AI globally, failing to address the challenges of AI integration in Pakistan risks widening the global digital divide in education. Therefore, this study fills a critical gap in the literature and informs policymakers, curriculum designers, and teacher training institutions about effective and context-sensitive strategies for AI-enhanced ESL instruction.

This study seeks to answer the following research questions:

1. How do ESL teachers in Pakistan integrate AI tools into their instructional practices?

2. How does AI integration influence students' vocabulary acquisition and writing proficiency in Pakistani ESL classrooms?

3. What challenges do teachers encounter in implementing AI-assisted instruction, and how do these challenges be addressed to maximize AI's potential in ESL education?

## Literature review

Integrating Artificial Intelligence (AI) into English as a Second Language (ESL) education has become a prominent area of research and practice, particularly in contexts marked by resource limitations, such as Pakistan. To critically examine AI's pedagogical potential and challenges, this study draws upon three interrelated theoretical perspectives: constructivist learning theory, sociocultural theory, and adaptive learning theory. These frameworks provide a multifaceted understanding of how AI technologies can facilitate or hinder language acquisition and the role of cultural and contextual factors in shaping effective AI integration.

Constructivist Learning Theory posits that learners actively construct knowledge through engagement with their environment rather than passively receiving information [11,12]. This theory emphasizes meaningful interaction, exploration, and reflection as core processes in learning. In ESL education, constructivism emphasizes the importance of interactive and adaptive learning experiences tailored to individual learners' prior knowledge and skills. AI-based language learning tools embody constructivist principles by providing personalized learning pathways, real-time feedback, and adaptive scaffolding that adjust to the learner's evolving competencies [2]. For example, AI systems can dynamically modify task complexity and suggest tailored exercises that align with learners' performance patterns. Such features foster autonomous learning and support students in refining their linguistic abilities through iterative practice. However, constructivist theory also stresses the critical role of scaffolding, where more knowledgeable agents, typically teachers or peers, provide support that enables learners to achieve tasks beyond their immediate capabilities [13]. While AI tools offer automated feedback and progressive difficulty adjustments, they often lack the nuanced, contextual understanding necessary to fully replace human scaffolding. This limitation is particularly salient in Pakistan's ESL classrooms, where teacher-mediated guidance remains essential for addressing complex linguistic challenges and motivational needs. Consequently, effective AI integration should be conceptualized as complementary to, rather than substitutive of, teacher facilitation.

Sociocultural Theory, formulated by Vygotsky [12], extends this perspective by emphasizing language learning as a socially mediated process embedded within cultural contexts. It foregrounds collaborative learning, peer interaction, and the co-construction of knowledge through dialogue. AI technologies have the potential to enhance social learning by enabling virtual collaboration, peer review, and interaction with AI-driven conversational agents [6]. These affordances align with the theory's emphasis on mediated interaction and socially situated learning. Nonetheless, the individualized nature of many AI applications can inadvertently reduce opportunities for face-to-face communication and peer collaboration, which are culturally valued in Pakistani classrooms. Moreover, sociocultural theory draws attention to the cultural relevance of learning materials. Content that resonates with learners' sociocultural backgrounds enhances engagement and comprehension [14]. Many existing AI platforms, however, rely heavily on globally standardized English materials that lack local cultural contextualization. This mismatch poses challenges for Pakistani ESL

learners, who may locate such content less engaging or meaningful. Addressing this gap requires the development of AI tools that are sensitive to the linguistic and cultural nuances of Pakistani learners, thereby fostering more authentic and effective language learning experiences.

Adaptive Learning Theory integrates insights from constructivist and sociocultural paradigms, emphasizing the importance of tailoring instruction to meet individual learner needs. AI-driven adaptive learning systems analyze learners' real-time performance data to adjust content, pace, and difficulty, optimizing the learning trajectory [1]. This responsiveness is especially valuable in ESL contexts, where learners vary widely in proficiency, learning styles, and educational backgrounds. By providing targeted interventions for specific linguistic difficulties such as grammar, vocabulary, and pronunciation, AI tools align well with adaptive learning principles that advocate for personalized instruction and self-paced progress [2]. Despite these strengths, adaptive learning theory also highlights the irreplaceable role of educators in addressing complex linguistic subtleties and sustaining learner motivation [5]. Particularly in Pakistan, where educational practices are often teacher-centered and motivational support is crucial, the integration of AI requires careful balance. Teachers' expertise in contextualizing feedback, fostering learner autonomy, and providing emotional encouragement remains essential to complement AI's automated functionalities. This interplay between technological and human instruction is a focal point of the present study.

Globally, AI applications have demonstrated promising impacts on language learning through automated grading, personalized feedback, and adaptive learning environments [15]. Automated grading systems alleviate assessment burdens in large classrooms, providing timely and objective feedback that supports learner development. However, research also highlights persistent challenges: AI struggles with linguistic nuances, including idiomatic expressions, pragmatics, and culturally specific language use [5,16]. Moreover, AI-generated feedback often lacks the depth and contextual relevance that experienced human instructors provide, particularly in advanced writing tasks requiring critical judgment.

In the Pakistani ESL context, these challenges are compounded by infrastructural and pedagogical barriers. The absence of reliable digital infrastructure in rural and under-resourced areas often restricts equitable access to AI tools [4]. Additionally, many teachers report insufficient training and confidence in employing AI-based pedagogies [17]. Despite these obstacles, AI holds potential for addressing persistent linguistic challenges in Pakistan. Vocabulary enhancement tools, for instance, have been shown to improve lexical acquisition when combined with teacher support [18]. Similarly, AI-assisted writing platforms facilitate iterative revision, though their effectiveness depends heavily on contextual guidance from instructors [5].

Situating AI integration within constructivist, sociocultural, and adaptive learning theories enables a detailed understanding of its pedagogical affordances and constraints. While AI supports personalized, adaptive, and socially mediated learning, its effectiveness is contingent upon culturally relevant content, infrastructural accessibility, and the continued involvement of skilled teachers. This study contributes to the growing discourse on AI in ESL education by critically examining these dimensions within the Pakistani context, offering insights for educators, policymakers, and developers aiming to optimize AI-enhanced language learning.

While global research offers fundamental explanations for AI's pedagogical affordances, it is equally important to examine how these tools are understood and applied within South Asian educational settings, where linguistic diversity and resource limitations create particular obstacles. In the South Asian region, recent scholarship has begun to explore the intersection of AI and language education, albeit at a nascent stage. For instance, Memon et al. [19] examined the role of AI-powered language learning tools in enhancing English communication skills among Pakistani learners, highlighting improved learner engagement but also flagging infrastructure gaps. Similarly, Hussain et al. [20] emphasized the transformative potential of AI in Pakistani classrooms but warned against overdependence on imported models that disregard local pedagogical practices. Regional studies in India and Bangladesh [21] have echoed similar concerns, stressing the need for AI systems to accommodate multilingual learning environments and culturally embedded pedagogies. These findings underscore the importance of localizing AI integration frameworks to reflect sociolinguistic diversity and educational

inequities prevalent in South Asian contexts. This study builds on and extends these regional insights by offering empirical evidence from Pakistani ESL classrooms using both qualitative and quantitative methods.

## Research design

The current study employed an exploratory sequential mixed-methods design, which involves collecting and analyzing qualitative data first, followed by a quantitative phase to validate and expand on the findings [22]. This design was chosen to comprehensively explore the integration of AI tools in Pakistani ESL classrooms, capturing both the depth of participants' experiences and the breadth of AI's impact on learning outcomes. The study adopts a pragmatic paradigm, prioritizing practical solutions to the research problem and leveraging the strengths of both qualitative and quantitative methods to provide a holistic understanding of AI integration in ESL education [23]. From an epistemological perspective, the study is grounded in pragmatism, which emphasizes the practical application of research and the use of multiple methods to address complex problems [24]. From an ontological perspective, the study recognises that the sociocultural and infrastructural contexts of Pakistani ESL classrooms shape multiple realities. This perspective aligns with the study's aim to explore how AI tools are perceived and utilized in a specific, resource-constrained environment, where participants' experiences may differ significantly from those in more technologically advanced settings.

The exploratory sequential design was selected for three key reasons. First, the qualitative phase (classroom observations and focus group discussions) allowed for an in-depth exploration of teachers' and students' experiences with AI tools, identifying challenges, benefits, and perceptions unique to the Pakistani context. These insights informed the development of the quantitative phase (pre- and post-tests), which measured the impact of AI tools on students' writing and vocabulary skills. The sequential approach grounded the quantitative measures in the participants' realities, thereby enhancing the validity of the findings. Second, the mixed-methods design addresses the complexity of the research problem, which involves both subjective experiences (e.g., perceptions of AI tools) and objective outcomes (e.g., improvements in language proficiency). By combining qualitative and quantitative data, the study captures the complexities of AI integration, balancing rich, contextual insights with measurable evidence of its impact. Finally, the design helps address gaps in the literature, which often focuses on either the pedagogical affordances of AI tools or the infrastructural challenges of their implementation but rarely integrates both perspectives. This study bridges this gap by providing a comprehensive understanding of how AI tools can be effectively integrated into ESL education, particularly in resource-constrained contexts like Pakistan. The findings offer practical recommendations for policymakers and educators, ensuring they are both theoretically grounded and actionable.

## Sampling techniques

To align with the exploratory sequential mixed-methods approach, this study employed different sampling techniques during the qualitative and quantitative phases, ensuring both depth and breadth in data collection. In the initial qualitative phase, which involved classroom observations and focus group discussions (FGDs), purposive sampling was used to identify information-rich participants. Specifically, five ESL teachers (three females and two males), each holding MS degrees in Applied Linguistics and trained in AI integration by the Higher Education Commission (HEC) of Pakistan, were selected. These teachers had prior exposure to AI tools and were actively engaged in implementing them in their instructional practices. Students selected for qualitative observation represented diverse socioeconomic and linguistic backgrounds, reflecting the multilingual and stratified educational context of Pakistan. This purposive approach ensured a comprehensive understanding of classroom dynamics, teacher experiences, and contextual challenges related to AI integration.

In the subsequent quantitative phase, convenience sampling was employed to select 100 third-semester ESL students from a private university. The inclusion criterion of a minimum CGPA of 3.5 was established to ensure baseline academic proficiency and to select participants who were sufficiently prepared to engage meaningfully with AI-assisted instructional

interventions. This criterion aimed to minimize variability caused by differing academic capabilities, allowing for a clearer assessment of AI's pedagogical effects. However, it is acknowledged that this selective threshold may introduce sampling bias, limiting the generalizability of findings to the broader population of ESL learners, including those with lower academic performance. To mitigate this limitation, demographic variables such as gender, socioeconomic status, and linguistic background were balanced within the sample to reflect diverse learner profiles as much as possible. Additionally, the mixed-methods design incorporating qualitative data helped provide a richer understanding of AI's impact beyond purely academic metrics. Future research should consider including a wider range of academic achievement levels to enhance the representativeness and applicability of results across different learner groups. These 100 students were then randomly assigned to either the experimental group (AI-assisted instruction) or the control group (traditional instruction), with 50 students in each group. While convenience sampling and the CGPA cutoff limit broad generalizability due to non-random and selective participant inclusion, the two-stage sampling strategy allowed the researchers to capture in-depth qualitative insights to inform the design of valid and contextually grounded quantitative instruments, enabling a comparative evaluation of AI's impact on student performance.

## Instruments

The data collection process in this study employed a multi-method approach, incorporating both quantitative and qualitative instruments to ensure depth, consistency, and validity. The instruments included a structured observation protocol, focus group discussion (FGD) guides, and separate pre- and post-tests designed to assess students' vocabulary acquisition and academic writing skills. The vocabulary and writing assessments were implemented as two distinct tests, rather than a single integrated test, to enable focused evaluation of each skill area. The Vocabulary Pre-Test and Post-Test consisted of multiple-choice, contextual usage, and short-definition items aligned with the ESL curriculum (see S1 File). The Writing Pre-Test and Post-Test included a rhetorical analysis task and an argumentative essay prompt, both structured to measure higher-order writing competencies. These assessments were administered at two time points, at the start (Week 1) and the end (Week 16) of the intervention period—to both the experimental and control groups (N = 100, 50 per group). The experimental group received AI-assisted instruction, while the control group was taught using conventional methods without AI integration. All assessments were conducted in controlled classroom settings immediately following regular instruction to maintain uniformity in testing conditions. To ensure content validity, both vocabulary and writing instruments were reviewed by subject-matter experts in applied linguistics. A pilot test was conducted with a sample of students (not included in the main study) to identify ambiguities and ensure clarity. The reliability of the tests was confirmed through Cronbach's alpha ($\alpha = .81$), indicating high internal consistency.

In addition to testing, classroom observations were carried out using a structured observation protocol (see S1 File) that documented teacher practices, student engagement, and the use of AI tools. To reduce observer bias, multiple researchers independently recorded observations, which were then cross-verified for consistency. Furthermore, qualitative insights were gathered through two Focus Group Discussions (FGDs) conducted with all five participating teachers (see S1 File). Each FGD lasted approximately 60 minutes and was led by two trained moderators who ensured balanced participation and minimized dominance by individual voices. The FGD guide included open-ended prompts exploring teacher perceptions of AI integration. Discussions were recorded, transcribed verbatim, and analyzed thematically to supplement the quantitative findings. This triangulated data collection strategy, integrating classroom observations, focus group discussions, and experimental assessments of student performance, enhanced the credibility and trustworthiness of the findings while ensuring methodological rigor and alignment with the study's objectives.

## Data collection procedures

After obtaining ethical approval and necessary permissions, written informed consent sheets were distributed among students and teachers. After seeking their consent, the researcher collected the data. In line with ethical research

practices, strict measures were implemented to ensure participant privacy and data protection. All video recordings were securely stored in encrypted, access-controlled digital drives and were used exclusively for research purposes. Students and teachers were clearly informed that participation was voluntary and that they could withdraw at any stage without penalty. Video recordings were anonymized during transcription, with identifying details removed or pseudonymized. Additionally, consent forms detailed how data would be stored, shared, and disposed of, ensuring transparency. Institutional review board (IRB) approval was secured before commencing the study, and ethical guidelines from the Higher Education Commission of Pakistan were rigorously followed. The study started on Monday, 1st January 2024, and ended on July 7, 2024. Researchers conducted natural classroom observations to study the integration of AI tools in real-time. Detailed field notes and video recordings documented verbal and non-verbal interactions, classroom settings, and participant behavior. To minimize observer bias and ensure reliability, multiple researchers cross-verified the findings. Participants' anonymity was preserved throughout. During a 16-week observation period, 32 classroom sessions were systematically observed, each lasting 90 minutes and conducted twice a week in a controlled environment. Multiple observers independently recorded their observations, which were then cross verified for consistency. Structured focus group discussions with teachers followed these observations, providing deeper insights into their experiences and perceptions of AI integration. Discussions were recorded, transcribed, and analyzed thematically, with triangulation and peer debriefing enhancing reliability. In addition to qualitative methods, quantitative data was collected through pre-test and post-test instruments designed to measure students' writing and vocabulary skills aligned with ESL curricula and standardized assessment practices. A pilot phase was conducted to refine the instruments, and data was analyzed using robust statistical methods to address potential biases and provide a comprehensive evaluation of AI's impact on learning outcomes.

## Data analysis procedures

Thematic analysis of qualitative data from natural observations and focus group discussions (FGDs) was conducted using Atlas.ti software, following Braun and Clarke's six-step framework [25]. The process began with familiarization with the data, where researchers reviewed observation notes and FGD transcripts multiple times to gain a deep understanding of the content (see Table 1). Next, initial codes were generated by systematically labeling meaningful segments of the data. These codes were then organized into potential themes, which were reviewed and refined to ensure they accurately

Table 1. Thematic analysis process with illustrative data, codes, and emerging themes.

| Step | Description | Example from data | Initial codes | Refined themes |
|---|---|---|---|---|
| **1. Familiarization** | Researchers reviewed observation notes and FGD transcripts multiple times. | Teacher expressed difficulty using AI tools due to lack of training. | Teacher challenges with AI tools | Pedagogical Challenges |
| **2. Generating Codes** | Meaningful segments of data were labeled with descriptive codes. | Students reported enjoying personalized exercises but missed peer interaction. | Student engagement with AI, Peer isolation | Student Engagement, Social Learning |
| **3. Identifying Themes** | Codes were grouped into potential themes based on patterns and relationships. | Teachers struggled to balance AI tools with traditional teaching methods. | Balancing AI and traditional methods | Pedagogical Integration |
| **4. Reviewing Themes** | Themes were reviewed to ensure they accurately represented the data. | Students found AI feedback helpful but sometimes irrelevant to advanced writing tasks. | AI feedback relevance, Writing challenges | Feedback Quality, Writing Development |
| **5. Defining Themes** | Final themes were defined and named, capturing key insights. | Teachers emphasized the need for localized AI tools to address cultural nuances. | Localization of AI tools, Cultural relevance | Localized AI Solutions |
| **6. Reporting** | Themes were used to structure the findings and discussion sections. | AI tools improved vocabulary retention but required teacher reinforcement. | Vocabulary retention, Teacher reinforcement | Vocabulary Development, Teacher Role |

represented the data. The final themes were defined and named, capturing the key patterns and insights related to AI integration in ESL classrooms.

To illustrate the analytical process, Table 1 provides an example of how codes were developed and refined into themes. For instance, initial codes such as "teacher challenges with AI tools" and "student engagement with AI" were grouped under broader themes like "Pedagogical Challenges" and "Student Engagement." This step-by-step approach ensured transparency and rigor in the thematic analysis. Triangulation and peer reviews ensured credibility and dependability. Triangulation involved cross-verifying findings from multiple data sources (observations, FGDs, and pre/post-tests) and perspectives (teachers and students). Peer reviews were conducted by having two independent researchers analyze a subset of the data and compare their findings to ensure consistency and reduce bias. Discrepancies were resolved through discussion, further enhancing the reliability of the analysis. For quantitative data, statistical analysis was performed using SPSS (see S2 File). Descriptive statistics, including means and standard deviations, gave an explanation for students' performance, while the reliability of pre- and post-tests was validated using Cronbach's alpha. Inferential statistics, specifically paired-sample t-tests, were employed to compare pre- and post-test results, validating the observed effects. By integrating these qualitative and quantitative methods, the study ensured a robust and comprehensive analysis of the data, mitigating potential biases and enhancing the reliability of findings.

## Results

This section presents the findings of the study in alignment with the three research questions outlined earlier in the introduction. Each subsection corresponds to one research question and systematically reports the outcomes derived from both the qualitative and quantitative phases of the study. The integration of these data sources allows for a comprehensive understanding of the impact of AI-assisted language learning on ESL students' academic performance, perceptions, and classroom experiences.

### Integration of AI tools into ESL instructional practices in Pakistan

The study reveals distinct findings from each methodological approach while addressing the first research question about how AI tools are integrated into ESL instructional practices in Pakistan:

The findings from classroom observations highlight that teachers adopt structured strategies to incorporate AI technologies, primarily using them to enhance writing proficiency, support vocabulary development, and facilitate individualized learning. Observations demonstrated that teachers integrate AI-powered writing aids such as Grammarly and QuillBot into peer-review tasks, guided writing activities, and automated grammar correction exercises.

The findings from focus group discussions reveal important traits about these implementations. Teachers reported that while these tools provide instant feedback on grammar, punctuation, and style, they sometimes fail to interpret context accurately. For example, participants shared that culturally specific references, including mentions of Pakistani traditions, were incorrectly marked as errors despite being grammatically accurate. Teachers in the focus groups emphasized that this limitation points to the need for AI systems that can recognize and accommodate local linguistic and cultural expressions.

For vocabulary instruction, classroom observations indicate that teachers use platforms like Quizlet and Memrise to introduce new words through interactive flashcards, spaced repetition, and game-based quizzes. However, the findings from focus group discussions with teachers indicate that long-term retention often required follow-up activities. One instructor explained that while AI applications can present vocabulary in engaging formats, learners still need teacher-guided exercises and contextual use to fully grasp new terms.

The findings from classroom observations of personalized learning demonstrate that intelligent tutoring systems such as Duolingo and Carnegie Learning adjust exercise difficulty based on students' performance. However, focus group

discussions reveal concerns from teachers about this approach. Teachers expressed apprehension that extensive reliance on self-paced AI tools might reduce peer interaction opportunities, while students remarked that although they benefited from customized instruction, they missed the exchange of ideas during group work.

Regarding implementation challenges, the findings from focus group discussions identify several key obstacles. Technical challenges such as unreliable internet access and outdated devices were frequently mentioned as disrupting lessons. Differences in teachers' digital skills emerged as another barrier, with some educators facing difficulties due to limited training. The discussions also revealed resistance from some teachers who view AI as threatening established practices rather than supporting them.

The findings from focus group discussions also highlight ethical concerns, particularly regarding data protection and algorithmic fairness. Teachers raised significant concerns about student privacy and stressed the importance of compliance with local data regulations. Participants noted that some AI systems prioritize Western English norms, often flagging accurate local usage as incorrect, which can discourage learners and reinforce linguistic inequality.

## The influence of AI integration on vocabulary acquisition and writing proficiency in Pakistani ESL classrooms

This study provides compelling evidence that incorporating intelligent learning platforms markedly enhances vocabulary acquisition and writing proficiency for learners in Pakistani ESL classrooms, addressing the second research question about the influence of AI integration. Employing a rigorously designed experimental framework comparing conventional instruction with technology-supported learning, the research reveals substantial improvements across key language competencies. These results illuminate how the targeted use of digital tools can reshape English language instruction while drawing attention to essential factors that condition their effectiveness.

Quantitative findings indicated a notable improvement in vocabulary learning among students who engaged with adaptive platforms such as Quizlet and Memrise. Performance data from vocabulary assessments suggests that these tools contributed to measurable learning gains. The pre- and post-test vocabulary scores revealed statistically significant improvements for both the experimental and control groups. The experimental group exhibited a greater improvement compared to the control group, with a mean pre-test score of 12.34 rising to 17.89 in the post-test gain of approximately 45%. A paired-samples t-test confirmed this difference as statistically significant, $t(98) = -21.47$, $p < .001$. The corresponding effect size, Cohen's $d = 1.12$, falls within the conventionally large range (Cohen, 1988), indicating a strong impact of the intervention. In comparison, the control group improved from a mean of 12.41 to 14.12 (a 14% gain), $t(98) = -8.32$, $p < .001$, with an effect size of $d = 0.67$, as shown in Table 2.

These findings suggest that adaptive learning platforms offering features such as tailored word banks, spaced repetition, and gamified quizzes can enhance vocabulary acquisition when integrated into classroom instruction. Nevertheless, potential threats to internal validity, such as test reactivity and participant motivation effects, warrant cautious interpretation. While efforts were made to control these factors through standardized testing conditions, their influence cannot be entirely ruled out.

Qualitative data from teacher observations and focus group discussions provided important contextual insights. Teachers noted that while digital tools effectively introduced and reinforced vocabulary, guided practice, and teacher-led

**Table 2. Pre- and post-test vocabulary scores for experimental and control groups.**

| Group | Pre-test | Post-test | Gain | t-test | p-value | Cohen's d |
|---|---|---|---|---|---|---|
| Experimental | 12.34 | 17.89 | +45% | −21.47 | <.001 | 1.12 |
| Control | 12.41 | 14.12 | +14% | −8.32 | <.001 | 0.67 |

*Note. Values represent mean scores (SDs in parentheses). N = 50 per group. df = 98 for all tests.*

scaffolding remained essential for promoting long-term retention and meaningful integration into students' communicative abilities. These insights underscore the complementary role of adaptive platforms in supporting, rather than substituting, direct instruction.

In the domain of writing, tools such as Grammarly and QuillBot facilitated iterative drafting and revision processes by providing automated feedback on grammar, syntax, and coherence. The experimental group's writing scores improved from a pre-test mean of 10.56 to a post-test mean of 15.42 (+46%), t(98) = −18.93, p < .001, with an effect size of d = 1.03. The control group demonstrated a smaller gain, from 10.62 to 12.03 (+13%), t(98) = −7.85, p < .001, with an effect size of d = 0.64 (see Table 3).

Although both groups showed significant improvements, the larger gains in the experimental group demonstrate the potential of AI-assisted tools to support writing development. However, teacher feedback revealed important limitations. Automated systems occasionally flagged culturally specific expressions or regional lexical items as errors, reflecting a bias toward standardized English norms. Such instances risk undermining learner confidence and emphasize the continuing need for teacher intervention, especially in argumentation, rhetorical structuring, and cultural appropriateness.

Implementation outcomes also varied based on contextual factors. Schools with strong technological infrastructure and targeted professional development for teachers saw more pronounced learning gains. Conversely, disparities in digital access, particularly between urban and rural settings, contributed to unequal learning opportunities. These findings highlight the importance of embedding digital tools within a thoughtfully designed pedagogical framework, supported by adequate teacher training and equitable access to resources.

Thus, the observed improvements in vocabulary and writing, accompanied by large effect sizes, suggest that when used strategically, adaptive platforms and AI-assisted writing tools can effectively complement traditional language instruction. At the same time, these results emphasize the need for context-aware implementation, ongoing teacher involvement, and critical attention to validity considerations in future research and practice.

## Challenges and solutions in implementing AI-assisted instruction in ESL education

This study finds that teachers in Pakistani classrooms face several significant barriers that hinder the effective integration of AI-assisted instruction in ESL education, addressing the third research question about the challenges and potential solutions related to this implementation. Data obtained through FGDs and classroom observations reveal that one primary obstacle is inadequate technological infrastructure, particularly in rural and under-resourced schools. Many educators report frequent disruptions due to unreliable internet connectivity, power outages, and outdated hardware that cannot support advanced AI applications. These technical limitations, observed consistently across multiple sites, force teachers to abandon AI tools mid-lesson and revert to traditional methods, resulting in inconsistent learning experiences. Furthermore, the digital divide between urban and rural institutions, as reported by teachers, exacerbates educational inequalities by depriving students in technologically underserved areas of AI's potential benefits.

Another critical challenge lies in teacher preparedness and professional development. Findings from classroom observation and focus group discussions indicate that many ESL instructors lack sufficient training to effectively incorporate AI tools into their pedagogical practices. This knowledge gap manifests in several ways: some teachers use AI applications

**Table 3. Pre- and post-test writing scores for experimental and control groups.**

| Group | Pre-test | Post-test | Gain | t-test | p-value | Cohen's d |
|---|---|---|---|---|---|---|
| Experimental | 10.56 | 15.42 | +46% | −18.93 | <.001 | 1.03 |
| Control | 10.62 | 12.03 | +13% | −7.85 | <.001 | 0.64 |

*Note. Scoring rubric assessed grammar, coherence, and task achievement. N = 50 per group.*

only for basic functions like grammar checking, while others struggle to interpret and contextualize AI-generated feedback for students. FGDs data also show that resistance to technological change persists among educators who perceive AI as a threat to traditional teaching methods or as compromising their professional autonomy. This skepticism, corroborated through qualitative FGDs, often stems from unfamiliarity with AI technologies and concerns about being replaced by automated systems rather than supported by them.

The pedagogical limitations of current AI systems present further implementation challenges. Document analysis of classroom AI tool usage and teacher-reported case studies reveal that most AI platforms are designed for Western educational contexts and often fail to accommodate the linguistic and cultural discrepancies of Pakistani English. Teachers report frequent instances where AI platforms incorrectly flag locally appropriate expressions as errors or offer suggestions that conflict with cultural norms. These biases, documented through classroom observation logs, create confusion for students and require additional teacher intervention to correct AI-generated feedback. Furthermore, qualitative insights indicate that AI systems currently focus predominantly on lower-order writing skills such as grammar and vocabulary, leaving higher-order competencies like critical thinking, argument development, and cultural relevance largely unsupported by technology.

Ethical concerns regarding data privacy and student surveillance also complicate AI integration. Teachers participating in FGDs express apprehension about how student data is collected, stored, and used by AI platforms, particularly when dealing with minors. The absence of clear policies and safeguards around educational data privacy echoes these concerns across multiple data sources, including classroom observation. Additionally, the proprietary nature of many AI algorithms, as noted in discussions with educators and technology specialists, makes it difficult for teachers to understand how assessments are generated or how to contest potentially biased outcomes.

To maximize AI's potential in ESL education, these challenges require targeted solutions at multiple levels. Infrastructure development must be prioritized, particularly in rural areas, through government initiatives and public-private partnerships that provide reliable internet access and modern devices, as suggested by participants and policy documents reviewed. Teacher training programs, based on the needs expressed during FGDs, should go beyond basic technical skills to develop educators' capacity for critically evaluating and effectively integrating AI tools within culturally responsive pedagogies. Professional development efforts must address both the practical use of technology and the philosophical shift toward augmented rather than automated teaching.

In addition, AI developers need to collaborate with local educators to create more culturally and linguistically appropriate systems. This recommendation is based on teacher feedback collected during the study. It includes training algorithms on diverse datasets that reflect Pakistani English variants and offers customization options that allow for sensitivity to local expressions. Ethical frameworks must also be established to ensure transparent data practices, with clear guidelines for student privacy and consent, an issue consistently raised in FGDs and policy analyses. Institutional policies should mandate regular audits of AI systems for bias and accuracy while maintaining human oversight in assessment processes.

Ultimately, successful implementation requires viewing AI as a pedagogical partner rather than a replacement. By addressing these challenges through coordinated efforts among policymakers, technologists, and educators, as indicated by cross-stakeholder perspectives gathered during the study, AI can realize its potential to enhance ESL instruction while respecting local contexts and preserving the essential human elements of language education. The path forward lies in developing sustainable, equitable, and pedagogically sound models of AI integration that empower both teachers and students in the language learning process.

## Discussion

This section addresses the findings of the study in light of the three research questions outlined in the introduction. It critically interprets the results by drawing connections between the qualitative and quantitative data and situating them within the broader context of existing literature. The discussion is structured around the key thematic areas emerging from

the study: [1] the perceived affordances and challenges of AI-based tools in ESL instruction, [2] the measurable impact of AI-assisted learning on student performance, and [3] the evolving roles of teachers and learners in AI-integrated classrooms. Each subsection integrates empirical evidence with theoretical insights to provide a broad overview of how AI is shaping language education in the Pakistani context.

## AI integration in Pakistani ESL classrooms

This study affirms and extends prior research on the pedagogical affordances of AI in English language education, particularly in the Pakistani ESL context. Consistent with Ahmed et al. [26,27], the findings indicate that Pakistani teachers strategically incorporate AI-powered writing aids and vocabulary platforms for formative feedback and vocabulary enhancement. Yet, distinct from earlier studies primarily situated in Western contexts [28,29], this research emphasizes the cultural and linguistic mismatches inherent in existing AI tools, particularly their inability to process regional language varieties and cultural references accurately. Such misalignments affect learner confidence and error perception, emphasizing the urgent need for AI systems attuned to localized English variants. This concern resonates with localization theories [30] and sociolinguistic models of language variation [31], which highlight the importance of adapting linguistic technologies to the socio-cultural contexts of diverse user populations.

The findings further contribute to the growing recognition that AI tools are most effective when integrated with teacher mediation. While adaptive platforms like Quizlet and Memrise support vocabulary learning through gamification, long-term retention and meaningful language use remain contingent on contextualized, teacher-led instruction, a dimension underexplored in prior quantitative studies [32,33]. This reinforces the sociocultural view of learning, particularly as articulated by Vygotsky [34], wherein AI serves as a complementary resource that supports, rather than replaces, teacher-led scaffolding and mediated learning processes.

Moreover, the study advances current debates on personalized instruction by highlighting a tension between adaptive learning and collaborative classroom dynamics. While tools like Duolingo and Carnegie Learning enable individualized pacing [35], their overuse risks diminishing peer interaction, a core component of language development. These findings support the case for hybrid instructional models that balance AI-driven customization with structured group engagement, aligning with collaborative learning theories such as social constructivism [34] and cooperative learning models [36].

The challenges identified— infrastructure deficits, variable digital literacy, and resistance to technology—align with observations in comparable contexts [37,38]. However, this study adds depth by illustrating how teachers' perceptions of AI as either a threat or a tool are shaped by affective and professional considerations. This insight underscores the need for culturally sensitive professional development that addresses both technical competence and teacher agency, drawing on models of reflective practice [39] and teacher agency frameworks [40] that emphasize empowerment, critical reflection, and adaptive expertise.

Ethical concerns, particularly data privacy and algorithmic bias, emerged as a critical theme. While these issues have been flagged in broader AI discourse [41,42], this study contextualizes them within Pakistani ESL classrooms, showing how such biases can reinforce linguistic hierarchies and undermine motivation. These findings call for AI design that embraces linguistic diversity and cultural inclusivity rather than perpetuating standardized norms, aligning with equity-focused educational technology frameworks [43] and scholarship on algorithmic bias [44], which advocate for inclusive, transparent, and culturally responsive AI systems.

Finally, the comparative analysis of experimental and control groups confirms the instructional value of AI-supported learning while validating the enduring effectiveness of conventional pedagogy. This underscores a blended instructional paradigm, wherein technology complements, but does not replace, teacher expertise, a model advocated by Alam and Mohanty [45] yet insufficiently explored in the Pakistani context. The present study thus offers a nuanced, contextually grounded perspective on AI integration in ESL education.

## AI integration and ESL proficiency in Pakistan

This study provides robust evidence that integrating adaptive learning platforms and intelligent writing tools enhances vocabulary acquisition and writing proficiency in Pakistani ESL classrooms. Aligning with Kem [46] and Chang and Sun [47], platforms like Quizlet and Memrise affirm the effectiveness of personalized, gamified learning environments. Distinctly, this study experimentally validates these effects within a South Asian context, addressing a notable gap as prior research [48,49] has predominantly focused on Western or East Asian populations.

Qualitative insights from teacher observations and focus groups enrich these findings by emphasizing the essential role of teacher facilitation in maximizing technology's benefits. This supports the socio-constructivist perspective [34] that learning is a socially mediated process, with digital tools serving as complements, not substitutes, for pedagogical guidance [50]. The blended instructional model promotes durable learning outcomes by combining AI feedback with teacher-led scaffolding.

Writing proficiency improvements resonate with prior research on automated writing evaluation (AWE) tools [51,52], highlighting the value of real-time, iterative editing enabled by applications such as Grammarly and QuillBot. Notably, this study identifies significant cultural and linguistic biases within these tools, consistent with Rana et al. [53], revealing the misclassification of region-specific lexical and syntactic features as errors. This limitation underscores the critical need for AI systems that are linguistically and culturally adaptive, reflecting sociolinguistic theories on language variation [31] and localization frameworks [30].

Beyond student outcomes, AI integration catalyzed a pedagogical shift toward more student-centered, facilitative teaching approaches. Teachers moved away from traditional lectures to guiding learners through AI-generated feedback, fostering metacognitive skills such as self-monitoring and critical reflection [39]. This transformation aligns with reflective practice theories [39] and underscores teacher agency as pivotal in navigating AI's affordances and limitations [40].

Challenges remain, particularly teachers' need for ongoing professional development to balance AI use without diminishing teacher-student rapport [54]. The findings suggest AI's greatest value lies in transforming instructional practices, supporting learner autonomy and deeper engagement, rather than merely improving test scores [55].

Institutional factors, including disparities in technological infrastructure and teacher training, critically mediated the success of AI integration. The observed urban-rural divide echoes digital equity frameworks [56], emphasizing the socio-economic and infrastructural variables that shape educational outcomes. This localized insight moves beyond global generalizations and offers targeted guidance for policymakers aiming to implement inclusive, context-sensitive digital learning strategies.

## Overcoming challenges in AI-driven ESL teaching

Regarding the third research question, the challenges and corresponding solutions in implementing AI-assisted instruction in ESL education, this study reaffirms the transformative potential of intelligent learning platforms in enhancing vocabulary acquisition and writing proficiency among Pakistani ESL learners. These results align with recent global research emphasizing the effectiveness of technology-assisted language learning [57,58], while also highlighting context-specific obstacles and the need for culturally responsive and pedagogically sound integration strategies. Unlike prior studies [59,60], predominantly conducted in Western or East Asian contexts, this research uniquely situates these technologies within the socio-cultural and infrastructural realities of Pakistani classrooms, providing fresh insights into both their potential and constraints in a South Asian educational setting.

Consistent with Fan and Zhang [61], who emphasize the importance of teacher mediation in technology integration, findings highlight that AI-supported tools alone are insufficient for sustained vocabulary retention and contextual language use. The blended approach combining adaptive digital platforms with skilled teacher facilitation emerges as a critical innovation, revealing how pedagogical expertise remains indispensable even amid rapid technological advances. This understanding advances existing literature by detailing specific teacher roles and interventions that optimize learning outcomes in resource-constrained environments, reinforcing sociocultural learning theories [34] that stress mediated learning.

Furthermore, the study's exploration of writing proficiency enhancement through platforms such as Grammarly and QuillBot echoes recent findings by Alharbi [62], who reported improvements in grammatical accuracy via automated feedback. However, this research distinctively contributes by highlighting the culturally rooted challenges of algorithmic bias against Pakistani English variants, an issue underexplored in prior work. By documenting teachers' corrective efforts to address these limitations, the study adds an important dimension to the discourse on localization of educational technologies [30] and calls attention to the need for culturally responsive AI systems, a concern deeply connected to sociolinguistic theories of language variation [31].

The challenges identified, including technological infrastructure gaps, limited teacher preparedness, and ethical concerns, align with those reported in regional studies [63,64], yet this study offers a comprehensive, empirically grounded perspective that integrates quantitative improvements with qualitative barriers. The delineation of rural-urban disparities in technology access contributes new empirical evidence of the digital divide's tangible impact on language learning equity in Pakistan, moving beyond theoretical assertions to concrete classroom realities, in line with digital equity frameworks [56].

Notably, this research advances the field by emphasizing the crucial role of teacher agency amid skepticism and resistance toward AI integration. While prior studies have acknowledged teacher attitudes [38], this study provides a detailed account of how fears surrounding professional autonomy influence pedagogical adoption, thereby enriching our understanding of human factors in educational technology implementation. Moreover, the study's call for collaborative development between AI designers and local educators to address linguistic and cultural mismatches breaks new ground. This practical recommendation, grounded in data, stresses the importance of algorithmic customization and ethical transparency, areas often overlooked in earlier investigations focusing narrowly on technical efficacy.

## Implications for policy and practice

The findings of this study offer several significant implications for educational policymakers and practitioners in Pakistan, particularly concerning the effective integration of AI tools in ESL instruction. First, there is a pressing need to develop comprehensive AI literacy training programs for both teachers and students. Many ESL instructors lack the necessary skills to critically evaluate and integrate AI tools into their pedagogical practices. Therefore, policy-level interventions must include structured training modules within teacher education programs to enhance digital pedagogical competencies. Such initiatives should also extend to learners, equipping them with the skills to effectively and ethically use AI tools for autonomous learning and academic communication.

Second, equitable access to digital infrastructure must be prioritized. Although this study focused on students from a private university with access to adequate technological resources, a large segment of ESL learners in public institutions remains digitally marginalized. National education policies must address this digital divide by investing in technological infrastructure, providing subsidized internet access, and incorporating AI-supported learning in public sector curricula to ensure that AI-enhanced education does not exacerbate existing inequalities.

Third, policymakers should consider the development of context-sensitive AI applications tailored to the sociolinguistic realities of Pakistani learners. The adoption of foreign-developed AI systems without localization may lead to incongruent feedback, cultural misalignment, and diminished pedagogical efficacy. Therefore, collaborative efforts between universities, local ed-tech firms, and language experts should be encouraged to develop AI systems aligned with national curriculum goals and regional linguistic practices.

Fourth, a critical yet often overlooked implication concerns the ethical deployment of AI in ESL education. While AI tools offer scalable and personalized learning opportunities, they also pose risks, particularly algorithmic bias and a lack of cultural adaptability. These limitations can compromise the validity of feedback and inadvertently reinforce linguistic imperialism or cultural insensitivity. To mitigate these risks, educational policies should mandate the use of AI tools trained on culturally and linguistically relevant data. Teacher education programs must also include modules that raise awareness about algorithmic limitations and train instructors to critically mediate AI feedback. Furthermore, public-private partnerships

can be incentivized to support the development of locally grounded, open-source AI tools that reflect the sociocultural and linguistic diversity of Pakistan. Such measures will ensure that the integration of AI in ESL instruction remains not only effective but also equitable, contextually appropriate, and ethically sound.

Finally, to address cultural misalignment in AI tools, the development of culturally responsive AI systems is essential. Such tools should be trained on corpora that include South Asian varieties of English, such as Pakistani English, to minimize misclassification of region-specific vocabulary and syntactic structures. Moreover, AI applications should allow users to toggle between standardized and localized English varieties. Developers are encouraged to integrate customizable settings that reflect the local curriculum and cultural norms, including the use of code-switching and culturally relevant idioms. Co-designing AI systems with local educators, curriculum developers, and linguists can help ensure linguistic inclusivity and pedagogical relevance. These improvements would not only increase tool accuracy but also foster learner confidence and engagement by validating their linguistic identities.

## Conclusion

This study set out to explore the integration of AI tools in Pakistani ESL classrooms, focusing on instructional practices, student outcomes in vocabulary and writing, and the challenges that hinder effective implementation. The findings offer a comprehensive picture of how AI is reshaping English language teaching in Pakistan while also revealing significant gaps and opportunities for future development. It was found that ESL teachers in Pakistan are increasingly incorporating AI tools into their teaching routines, albeit with varying levels of depth and confidence. Most teachers utilize these tools, such as Grammarly, QuillBot, and AI-based vocabulary apps, for lower-order language skills like grammar correction and vocabulary reinforcement. However, this integration often lacks pedagogical sophistication due to insufficient training. Teachers tend to adopt a tool-centric rather than learner-centered approach, limiting AI's potential for fostering critical thinking and contextual language use. Where AI integration has been most successful, it has occurred in blended environments where teachers actively mediate the learning process, customizing AI feedback and supporting student engagement. This highlights the indispensable role of human agency in navigating and enhancing AI-assisted instruction.

The study further found that AI integration has a measurable positive impact on students' vocabulary acquisition and writing proficiency. Students exposed to AI tools demonstrated improved accuracy in spelling, grammar, and word choice and displayed greater confidence in written expression. While these outcomes align with global research, this study highlights context-specific benefits emerging in localized learning environments when AI tools are strategically paired with teacher-led instruction. However, the gains in vocabulary and writing appear primarily in lower-order skills. Without deliberate teacher intervention, students seldom develop advanced writing competencies such as argumentative structure, coherence, and critical analysis. Hence, AI should be considered a supplementary resource rather than a substitute for comprehensive language instruction.

Moreover, the study identified four critical challenges: inadequate infrastructure, lack of teacher training, pedagogical limitations of AI tools, and ethical concerns. These barriers are particularly pronounced in rural and underfunded schools, exacerbating educational inequities. Teachers face technological disruptions, limited access to modern devices, and skepticism driven by concerns over job security and pedagogical autonomy. Additionally, current AI platforms often lack sensitivity to Pakistani English and cultural contexts, resulting in misleading feedback. Ethical considerations, especially regarding data privacy, further complicate AI adoption. To address these challenges, the study recommends multi-level interventions: infrastructural investment, culturally responsive AI development, teacher training that encompasses both technical and pedagogical dimensions, and the establishment of clear data governance frameworks.

Building on these findings, future research should focus on several key areas. First, there is a need for context-sensitive AI feedback systems that provide culturally and contextually relevant guidance, particularly for advanced writing tasks. Collaborative efforts between linguists, educators, and AI developers could refine algorithms to address the nuanced needs of ESL learners. Second, considering the challenges teachers face in integrating AI tools, targeted

professional development programs are essential. Future studies should investigate the long-term effects of such programs on teachers' confidence and competence in utilizing AI tools effectively. Third, balancing personalized learning with opportunities for peer interaction requires further exploration. Mixed-methods studies could shed light on the social dynamics of AI-supported classrooms and promote collaborative learning environments. Fourth, ethical and inclusive AI design should be prioritized, with research addressing data privacy, algorithmic biases, and the accommodation of linguistic diversity. Finally, longitudinal studies are needed to assess the sustained impact of AI tools on language learning outcomes, including retention, proficiency, and the long-term viability of AI-supported interventions.

## Limitations of the study

A key limitation of this study is its use of convenience sampling from a single private university, restricted to high-performing students (CGPA ≥ 3.5). This selective sampling introduces potential bias and limits the generalizability of the findings to the broader ESL learner population in Pakistan, particularly those in public institutions, rural areas, or with varying academic abilities. Additionally, the digital infrastructure at the study site may not reflect conditions in less resourced settings, further constraining external validity. While the study provides valuable insights into AI-assisted learning, future research should involve more diverse and representative samples, employ randomized sampling, and investigate long-term outcomes across different educational contexts.

## Supporting information

**S1 File. Appendices.**
(DOCX)

**S2 File. Research data.**
(RAR)

## Author contributions

**Conceptualization:** Tahir Saleem.

**Data curation:** Tahir Saleem, Muhammad Aslam.

**Formal analysis:** Tahir Saleem, Muhammad Aslam.

**Investigation:** Tahir Saleem.

**Methodology:** Tahir Saleem.

**Validation:** Aisha Saleem.

**Writing – original draft:** Tahir Saleem.

**Writing – review & editing:** Tahir Saleem, Aisha Saleem.

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
