## [Decision Letter · Decision Letter 0]

22 May 2025

PONE-D-25-13511Integrating AI in Pakistani ESL Classrooms: Teachers' Practices, Perspectives, and Impact on Student PerformancePLOS ONE

Dear Dr. Saleem,

Thank you for submitting your manuscript to PLOS ONE. After careful consideration, we feel that it has merit but does not fully meet PLOS ONE’s publication criteria as it currently stands. Therefore, we invite you to submit a revised version of the manuscript that addresses the points raised during the review process.

We look forward to receiving your revised manuscript.

Kind regards,

Mostafa Janebi Enayat, Ph.D.

Academic Editor

PLOS ONE

Journal Requirements:

3. Please include a copy of Table 1 which you refer to in your text on page 9.

Reviewers' comments:

Reviewer's Responses to Questions

**Comments to the Author**

1. Is the manuscript technically sound, and do the data support the conclusions?

Reviewer #1: Yes

Reviewer #2: Yes

Reviewer #3: Yes

Reviewer #4: Yes

2. Has the statistical analysis been performed appropriately and rigorously? 

Reviewer #1: Yes

Reviewer #2: Yes

Reviewer #3: Yes

Reviewer #4: Yes

3. Have the authors made all data underlying the findings in their manuscript fully available?

Reviewer #1: Yes

Reviewer #2: Yes

Reviewer #3: Yes

Reviewer #4: Yes

4. Is the manuscript presented in an intelligible fashion and written in standard English?

Reviewer #1: Yes

Reviewer #2: Yes

Reviewer #3: Yes

Reviewer #4: Yes

5. Review Comments to the Author

Reviewer #1: Your research offers insightful information about integrating AI into ESL classes in Pakistan. The evidence supporting AI's beneficial effects on writing abilities and vocabulary learning is strong. To further improve the use of AI in education, it will be essential to address the issues of infrastructure, cultural biases, and teacher training.

Reviewer #2: Areas for improvement:

1. Structure and Organization

a) Introduction: The introduction is quite long and contains a lot of theoretical information that could have been more effective if divided into its own section. A more explicit statement of purpose would have been useful to demonstrate the contribution of the study.

b) Literature Review: Some theories, such as constructivism and adaptive learning theory, are explained at length and are repetitive. A summary of the main points could have been more effective.

c) Methodology: The justification for selecting participants (only students with CGPA ≥ 3.5) needs to be clarified regarding possible bias in generalizing the findings.

d) Results and Discussion: Some sections of the discussion tend to repeat results without sufficient critical analysis in comparing with previous studies.

2. Arguments and Clarity of Contributions

a) The implications of the study are not sufficiently clearly stated regarding how education policy in Pakistan can more effectively adapt AI.

b) Criticisms of AI tools are mentioned, but not enough concrete solutions are offered to address issues such as algorithmic bias and lack of cultural sensitivity in AI feedback.

c) A more in-depth discussion of the impact of AI on teacher teaching methods is needed to show how AI actually impacts learning strategies, not just student outcomes.

3. Data Presentation and Clarity of Tables

a) Some of the statistical tables are quite dense but could be made more intuitive with the addition of clearer interpretations.

b) The use of visualizations (e.g., graphs) can help readers understand the impact of AI on the control vs. experimental groups more quickly.

Reviewer #3: A. Abstract Analysis

The abstract is good in explaining the problem, methodology, results, and final conclusion. At the end of the conclusion, the research findings and the expected follow-up are stated.

Suggestion: can be more explicitly strengthened regarding 1) the urgency of the research, especially in the context of theory or practical problems faced; 2) The implementation of the research stage from qualitative data collection techniques (interviews, observations) followed by quantitative analysis methods; 3) Mention the results of quantitative data to strengthen the results obtained;

B. Introduction Analysis

Advantages & Disadvantages

1) The background of this study is quite clear in describing the importance of the phenomenon being studied, especially in the context of the exploratory sequential mixed-methods method; 2) The author shows the reasons why this study is needed, but it could be strengthened by providing more empirical data or concrete examples that support the urgency of this study; 3) The article cites several related studies but does not analyze the gap between previous studies and this study; 4) If there is a gap between research and previous studies, it needs to be emphasized more explicitly to show the contribution of this study compared to previous studies; 5) The research question is explicit by stating 3 research objectives; 4) There are references to previous studies, but it needs to be more analytical.

Suggestions:

1) Can be strengthened by adding supporting data that shows the urgency of this research problem; 2) Analyze the results of previous research more. If the purpose of this research is to fill the gap in previous research, this section can be strengthened by stating the shortcomings or limitations of previous research more explicitly. 3) If necessary, add supporting references and compare more clearly how this research differs from previous studies or to what extent the implications of this research are practical or theoretical; 4) It is better to explain the weaknesses or limitations of previous research that make this research very necessary.

C. Methodology Analysis

This study has procedurally carried out the exploratory sequential design stages well. From the methodology, it can be seen that this study uses the results of qualitative analysis to compile quantitative instruments. The process of developing the instrument from the results of interviews or qualitative findings is converted into indicators in a quantitative questionnaire that is tested by expert validity on its quantitative instruments. It's just that it has not been explicitly explained what sampling techniques are used at each stage with the limitations of the sampling techniques used.

Suggestion: Explain the sampling technique used.

D. Analysis of Results and Discussion

The results presented reflect two stages in exploratory sequential mixed methods, namely the results of the qualitative and quantitative stages.

Advantages & Disadvantages

1) This study has a clear objective, in the form of 3 explicit research questions, but in the discussion of the writing, it is not structured how the results of this study answer each research objective explicitly. 2) The interpretation of the table data has not been explained explicitly in the discussion description; 3) This article connects several research results with previous findings but is not sufficient in analyzing the differences or similarities with previous research or the implications of the research results. 4) There are references to supporting theories, but they are not developed enough to clarify the theoretical implications of this study.

Suggestions:

1) The writing structure should be clearer in connecting each research result with the research question. 2) Add data interpretation for each table so that readers can understand the relevance of qualitative and quantitative data from the research results. 3) Discussion can highlight more what is new or unique about this research compared to previous research to show the contribution of this research. 4) Discuss whether the results of this research confirm or challenge existing theories and what the practical and theoretical implications of the results of this research are more explicitly.

E. Conclusion Analysis

Advantages & Disadvantages

Most of the conclusions have answered the research questions, but: 1) The writing structure is not explicit yet in connecting each conclusion of the research results with the previously formulated research objectives, and 2) The research findings are quite clear; there are recommendations for further research that can be carried out based on these findings.

Suggestion:

Be more structured in writing the conclusion so that the answers to each research question are clearly.

Reviewer #4: General Assessment

The manuscript examines the implementation of Artificial Intelligence (AI) in Pakistan, focusing on the opportunities and challenges faced in its adoption across various sectors. The topic is highly relevant given the global shift towards AI-driven solutions and their impact on economic and technological development. While the study provides meaningful insights, several areas require further development to improve clarity, methodological rigor, and coherence.

Strengths

1. Relevance: The study addresses a significant and timely topic, considering the growing role of AI in developing economies.

2. Structure: The manuscript follows a logical structure, making it easy to follow.

3. Literature Review: The discussion of previous studies is well-integrated, providing a solid foundation for understanding AI implementation in Pakistan.

4. Findings: The research highlights key opportunities and barriers to AI adoption, contributing to the discourse on digital transformation in emerging economies.

Areas for Improvement

1. Clarity and Writing Style

• Some sections contain lengthy and complex sentences that could be simplified for better readability.

• Minor grammatical and typographical errors should be corrected. A thorough proofreading or professional editing service is recommended.

2. Abstract

• The abstract mentions an improvement in specific skills but does not provide clear data on the extent of this improvement. It is essential to specify numerical or percentage-based data to substantiate the claim.

• The abstract summarizes the study's conclusions; therefore, since the paper addresses three key issues, the abstract should also be structured into three distinct sections, each with clear supporting data.

3. Methodology

• More details are needed regarding data collection. How was the data gathered, and what sources were used?

• If surveys or interviews were conducted, details about the sample size, participant selection, and questionnaire structure should be included.

• The study lacks a clear explanation of the criteria used to assess AI implementation.

• Ethical considerations related to data collection should be explicitly mentioned.

4. Data Analysis and Interpretation

• The statistical methods used should be explicitly stated. Were significance tests applied?

• Some conclusions appear speculative rather than data-driven. Ensure that all claims are backed by empirical evidence.

• The findings should be explicitly connected to the research questions or hypotheses.

5. Citations and References

• Some references appear outdated. Consider incorporating more recent studies to reflect the latest developments in AI implementation.

• A few citations in the text are missing corresponding references in the reference list. Ensure proper citation consistency.

Recommendation

Minor Revisions – The manuscript presents a valuable contribution, but revisions are needed to strengthen its methodology, clarity, and data analysis. The authors should address the issues outlined above before publication.

Additional Comments to the Author:

1. Consider elaborating on the specific AI applications in different sectors of Pakistan (e.g., healthcare, finance, education, and government).

2. The discussion section should explicitly compare the findings with previous studies to highlight the unique contributions of this research.

3. If possible, include real-world examples or case studies to illustrate the practical implications of AI implementation in Pakistan.

6. PLOS authors have the option to publish the peer review history of their article (what does this mean? ). If published, this will include your full peer review and any attached files.

**Do you want your identity to be public for this peer review?** For information about this choice, including consent withdrawal, please see our Privacy Policy .

Reviewer #1: No

Reviewer #2: **Yes: ** Ahmad Mustanir

Reviewer #3: No

Reviewer #4: No

---

## [Decision Letter · Decision Letter 1]

15 Jul 2025

PONE-D-25-13511R1Integrating AI in Pakistani ESL Classrooms: Teachers' Practices, Perspectives, and Impact on Student PerformancePLOS ONE

Dear Dr. Saleem,

Thank you for submitting your manuscript to PLOS ONE. After careful consideration, we feel that it has merit but does not fully meet PLOS ONE’s publication criteria as it currently stands. Therefore, we invite you to submit a revised version of the manuscript that addresses the points raised during the review process.

The reviewers have addressed a number of minor issues in your manuscript. Please apply them in the next revision. 

We look forward to receiving your revised manuscript.

Kind regards,

Mostafa Janebi Enayat, Ph.D.

Academic Editor

PLOS ONE

Journal Requirements:

Reviewers' comments:

Reviewer's Responses to Questions

**Comments to the Author**

1. If the authors have adequately addressed your comments raised in a previous round of review and you feel that this manuscript is now acceptable for publication, you may indicate that here to bypass the “Comments to the Author” section, enter your conflict of interest statement in the “Confidential to Editor” section, and submit your "Accept" recommendation.

Reviewer #1: All comments have been addressed

Reviewer #3: (No Response)

Reviewer #4: All comments have been addressed

2. Is the manuscript technically sound, and do the data support the conclusions?

Reviewer #1: Yes

Reviewer #3: Yes

Reviewer #4: Yes

3. Has the statistical analysis been performed appropriately and rigorously? 

Reviewer #1: N/A

Reviewer #3: Yes

Reviewer #4: Yes

4. Have the authors made all data underlying the findings in their manuscript fully available?

Reviewer #1: Yes

Reviewer #3: Yes

Reviewer #4: Yes

5. Is the manuscript presented in an intelligible fashion and written in standard English?

Reviewer #1: Yes

Reviewer #3: Yes

Reviewer #4: Yes

6. Review Comments to the Author

Reviewer #1: Using a suitable mixed-methods design, validated instruments, and thorough data analysis, the work is technically solid. The findings about AI's effects on writing and vocabulary are persuasively supported by the facts. There are no obvious issues with dual publication or research ethics, and ethical procedures were followed. The study has sound methodology.

Reviewer #3: A. Abstract

Done. But it would be better if the introduction in general was not omitted. (The integration of Artificial Intelligence (AI) in English as a Second Language (ESL) classrooms has gained prominence globally, yet its implementation in resource-constrained settings like Pakistan remains underexplored). If the abstract is considered too long, you can make the sentences more effective by still including the general background, research objectives, methodology, results, conclusions, and novelty.

B. Introduction

done

C. Methodologi

Done

D. Result and Discussion

Done

E. Conclusion

done

Reviewer #4: The manuscript titled "Integrating AI in Pakistani ESL Classrooms: Teachers' Practices, Perspectives, and Impact on Student Performance" presents a timely and relevant investigation into the use of artificial intelligence in English as a Second Language (ESL) education within a developing country context. The study is commendable for its comprehensive mixed-methods design, theoretical grounding in constructivist, sociocultural, and adaptive learning theories, and integration of both qualitative and quantitative data. It addresses a significant research gap by exploring AI integration in resource-constrained educational settings, an area that is often neglected in mainstream AI-in-education research, which typically focuses on technologically advanced countries. The use of classroom observations, focus group discussions, and pre- and post-testing offers a rich dataset, and the findings contribute valuable insights for both researchers and policymakers.

However, despite these strengths, the manuscript requires substantial revision before it can be considered for publication. First and foremost, the reported effect sizes for vocabulary and writing improvements (Cohen’s d = 4.12 and 3.67, respectively) are unusually large and statistically implausible. Such magnitudes of gain suggest either an overestimation or potential bias in measurement or sample selection. The authors should revisit their data analysis, clarify whether these values were computed correctly, and address any threats to internal validity, including the Hawthorne effect or test reactivity. Furthermore, the sample was drawn using convenience sampling from a single private university and was limited to high-performing students (CGPA ≥ 3.5). This restriction significantly limits the generalizability of the findings to the wider ESL learner population in Pakistan, particularly in rural or public educational institutions. Although the study briefly acknowledges this limitation, it should be discussed more explicitly, especially in the conclusion and implications sections.

In terms of structure and writing, the manuscript is generally well-organized but overly lengthy and at times repetitive, particularly in the discussion section. There is a tendency to restate the findings instead of synthesizing them with theoretical implications. The ethical considerations surrounding data collection, especially the handling of student video recordings and privacy protections, are underdeveloped and should be more clearly articulated. Additionally, while the manuscript rightfully critiques the cultural misalignment of Western-developed AI tools like Grammarly and QuillBot, it stops short of proposing practical solutions or suggesting design features for culturally responsive alternatives. This is a missed opportunity, given the study’s emphasis on contextual adaptation.

Moreover, the literature review could benefit from greater integration of regional and local studies to better situate the research within the South Asian context. Minor language issues are present throughout the manuscript, including occasional grammatical inconsistencies and awkward phrasing, which should be corrected through careful proofreading. The tables referenced in the text should be formatted and placed appropriately for reader clarity. The references are generally current and relevant, but consistency in citation formatting should be checked according to APA or the journal's required style.

In conclusion, the manuscript presents important and relevant findings, but revisions are needed to improve its methodological transparency, critical interpretation, and linguistic clarity. The potential for publication is strong, but only after these issues, especially the statistical anomalies and generalizability limitations, are thoroughly addressed.

7. PLOS authors have the option to publish the peer review history of their article (what does this mean? ). If published, this will include your full peer review and any attached files.

**Do you want your identity to be public for this peer review?** For information about this choice, including consent withdrawal, please see our Privacy Policy .

Reviewer #1: No

Reviewer #3: No

Reviewer #4: **Yes: ** Dr. Restu Januarty Hamid, S.Pd.I., M.Pd.

---

## [Decision Letter · Decision Letter 2]

10 Aug 2025

PONE-D-25-13511R2Integrating AI in Pakistani ESL Classrooms: Teachers' Practices, Perspectives, and Impact on Student PerformancePLOS ONE

Dear Dr. Saleem,

Thank you for submitting your manuscript to PLOS ONE. After careful consideration, we feel that it has merit but does not fully meet PLOS ONE’s publication criteria as it currently stands. Therefore, we invite you to submit a revised version of the manuscript that addresses the points raised during the review process.

While two reviewers have endorsed your revision, one reviewer has identified substantial concerns throughout the paper that require careful attention. Please address each of these comments thoroughly in the next version of your manuscript.

We look forward to receiving your revised manuscript.

Kind regards,

Mostafa Janebi Enayat, Ph.D.

Academic Editor

PLOS ONE

Journal Requirements:

Reviewers' comments:

Reviewer's Responses to Questions

**Comments to the Author**

1. If the authors have adequately addressed your comments raised in a previous round of review and you feel that this manuscript is now acceptable for publication, you may indicate that here to bypass the “Comments to the Author” section, enter your conflict of interest statement in the “Confidential to Editor” section, and submit your "Accept" recommendation.

Reviewer #1: All comments have been addressed

Reviewer #3: (No Response)

Reviewer #4: All comments have been addressed

2. Is the manuscript technically sound, and do the data support the conclusions?

Reviewer #1: Yes

Reviewer #3: Yes

Reviewer #4: Yes

3. Has the statistical analysis been performed appropriately and rigorously? 

Reviewer #1: Yes

Reviewer #3: Yes

Reviewer #4: Yes

4. Have the authors made all data underlying the findings in their manuscript fully available?

Reviewer #1: Yes

Reviewer #3: Yes

Reviewer #4: Yes

5. Is the manuscript presented in an intelligible fashion and written in standard English?

Reviewer #1: Yes

Reviewer #3: Yes

Reviewer #4: Yes

6. Review Comments to the Author

Reviewer #1: Thank you for sharing your manuscript. The use and incorporation of qualitative and quantitative data tells a comprehensive story about the use of AI in the ESL classrooms in Pakistan. The findings, especially those on teacher mediation, infrastructural AI-accompanied tool adaptation, and the cultural framing of AI in the educational ecosystem, are conceptually important and practically useful. The work has critical implications for practitioners and policy developers. I hope the study will meaningfully advance the conversation around context-sensitive and equitable AI use in language education.

Reviewer #3: Thank you to the author for the revisions to perfect this manuscript. In the previous review, I only requested improvements to the abstract. The abstract now meets IMRAD standards with effective sentences, making it less lengthy.

Reviewer #4: Suggestions For Improvement

ABSTRACT

Weakness:

• The abstract is dense and conceptually loaded, using technical terms like “thematic synthesis,” “three-dimensional engagement,” and “seven influencing factors” without clarification.

• It lacks a clear statement on the practical relevance of the findings (e.g., for teachers, curriculum developers, or LMS designers).

Suggestion:

• Simplify the language to make it more accessible to broader academic readers.

• Add 1–2 lines on the educational implications, e.g., how the findings can inform digital feedback practices in EFL classrooms or teacher development programs.

INTRODUCTION

Weakness:

• Several sentences are repetitive or wordy, making the argument harder to follow.

• The rationale for the study is underdeveloped; it lacks a strong contrast with previous systematic reviews or meta-analyses.

• The introduction does not distinguish between different digital feedback modalities (e.g., LMS-based, AI-generated, peer-reviewed) early enough.

Suggestion:

• Streamline the prose by removing redundancy and condensing long sentences.

• Clearly state how this study fills a gap in the literature (e.g., previous studies focused more on feedback quality, not engagement).

• Define key feedback environments (e.g., teacher, peer, automated) upfront to improve conceptual clarity.

LITERATURE REVIEW

Weakness:

• The literature review tends to be descriptive, summarizing prior studies without offering a critical evaluation of their methods, scope, or findings.

• The relationship between the three engagement dimensions (behavioral, cognitive, affective) is presented independently, without discussion of their interaction or overlap.

• Theoretical models (e.g., Ellis's or Han & Hyland’s frameworks) are not sufficiently compared or critiqued.

Suggestion:

• Include a critical synthesis of different conceptualizations of engagement, for example, how behavioral engagement supports cognitive development, or how affective responses influence uptake.

• Discuss tensions in the literature (e.g., disagreement over which type of engagement is most predictive of learning gains).

• Present a visual table or diagram comparing engagement frameworks from multiple scholars for clarity.

METHODOLOGY

Weakness:

• While the study follows PRISMA and offers a clear protocol, it does not report inter-rater reliability (e.g., Cohen’s Kappa) for article screening or coding, which limits replicability and trust in consistency.

• The exclusion of non-English and grey literature is only briefly acknowledged and not adequately justified.

• The search strategy lacks information on whether publication bias (e.g., overrepresentation of positive findings) was considered.

Suggestion:

• Report a reliability coefficient or percentage agreement for screening and coding to support methodological rigor.

• Justify the language and source restrictions with more clarity (e.g., limited access, need for peer-reviewed consistency).

• Include a statement on how publication bias was mitigated, such as through database diversity or reference chaining.

FINDINGS / THEMATIC ANALYSIS

Weakness:

• The discussion of cognitive engagement is in-depth, but affective and behavioral engagement are less developed. This creates an imbalance and may obscure how learners emotionally or behaviorally respond to feedback.

• The seven influencing factors are presented as discrete categories, but some overlap conceptually, such as “motivation” and “learner beliefs”.

• The interaction between types of feedback and types of engagement is not fully explored (e.g., do learners engage differently with peer vs teacher feedback?).

Suggestion:

• Expand the affective and behavioral sections with more examples, case comparisons, or quotes from studies.

• Clarify the boundaries and relationships between influencing factors, possibly using a visual concept map.

• Include a matrix or synthesis table mapping feedback types (teacher, peer, automated) to the three engagement types and seven influencing factors.

DISCUSSION

Weakness:

• The discussion does not deeply reflect on contradictions in the findings, for instance, situations where online feedback decreases rather than increases engagement.

• There is little attention to cross-cultural variation, even though EFL contexts vary widely in feedback acceptance and response.

• The link between the findings and larger theories of L2 acquisition or digital learning is underexplored.

Suggestion:

• Acknowledge and analyze inconsistencies or outliers in the data (e.g., where learners reject feedback or become disengaged).

• Discuss how cultural factors (e.g., power distance, educational norms) influence feedback engagement in EFL contexts.

• Connect the findings to broader learning theories such as sociocultural theory, self-determination theory, or engagement theory.

PEDAGOGICAL IMPLICATIONS

Weakness:

• The implications are general (e.g., “teachers should support learners’ engagement”) and do not offer concrete strategies for how this can be done.

• Does not account for technological challenges, like limited access to stable internet or platform usability.

Suggestion:

• Provide specific strategies for teachers, such as how to train students in feedback literacy, integrate reflection prompts, or use peer review rubrics.

• Include recommendations for platform designers (e.g., LMS developers) based on the findings, such as scaffolding, auto-feedback calibration, and feedback tracking.

• Discuss how to support engagement in low-resource digital environments.

LIMITATIONS AND FUTURE DIRECTIONS

Weakness:

• The limitations are acknowledged (e.g., language, database scope), but future research suggestions are too broad (“more studies needed”).

• No discussion on innovative methods for future study (e.g., emotion analysis, eye-tracking, or engagement analytics).

Suggestion:

• Recommend specific research designs (e.g., longitudinal mixed-methods, ethnographic studies, classroom interventions).

• Highlight unexplored populations (e.g., young learners, students with digital disabilities, rural EFL learners).

• Suggest integrating affective computing tools or learning analytics dashboards to track engagement.

CONCLUSION

Weakness:

• The conclusion summarizes findings but doesn’t offer a strong final insight or unified model.

• Contributions to theory vs practice are not explicitly distinguished.

Suggestion:

• End with a strong proposed framework or model (even a simplified version) that connects feedback type, learner engagement, and influencing factors.

• Divide the conclusion into two parts: (1) Theoretical contributions, and (2) Practical implications.

Overall Assessment

• The article is well-structured and methodologically sound.

• The main areas for improvement are:

o Deepening the critical engagement with literature.

o Strengthening affective and behavioral analyses.

o Providing more practical, specific pedagogical insights.

o Offering a clearer, more synthesized conclusion or framework.

7. PLOS authors have the option to publish the peer review history of their article (what does this mean? ). If published, this will include your full peer review and any attached files.

**Do you want your identity to be public for this peer review?** For information about this choice, including consent withdrawal, please see our Privacy Policy .

Reviewer #1: No

Reviewer #3: No

Reviewer #4: **Yes: ** Dr. Restu Januarty Hamid, S.Pd.I., M.Pd.

---

## [Author Response · Author response to Decision Letter 3]

19 Aug 2025

Dear Editor,

I am writing regarding the recent review report received for my manuscript entitled “Integrating AI in Pakistani ESL Classrooms: Teachers’ Practices, Perspectives, and Impact on Student Performance” .

All reviewers, with the exception of Reviewer 4, have recommended the publication of our manuscript. Upon thorough examination of Reviewer 4's feedback, it appears that the report pertains to a different study and does not accurately reflect the scope, methodology, or content of our submission. I respectfully present the following points for your consideration:

1. Terminology mismatch: The reviewer refers to “thematic synthesis,” “three-dimensional engagement,” and “seven influencing factors.” My study employs thematic analysis but not thematic synthesis, and no such engagement framework or “seven influencing factors” is used in my work.

2. Methodological mismatch: The report critiques the absence of inter-rater reliability (e.g., Cohen’s Kappa) for article screening, the exclusion of non-English and grey literature, and the lack of publication bias mitigation. These concerns are relevant to a systematic review or meta-analysis using PRISMA guidelines, whereas my study is an empirical mixed-methods classroom intervention involving AI-assisted ESL instruction in Pakistan.

3. Framework mismatch: The reviewer references “Ellis’s” and “Han & Hyland’s” frameworks, as well as a suggested matrix mapping feedback types (teacher, peer, automated) to engagement dimensions. My research does not investigate feedback-type comparisons or adopt these theoretical models; instead, it evaluates the pedagogical integration of AI tools such as Grammarly, QuillBot, Quizlet, and Memrise in actual classroom practice.

4. Scope mismatch: The comments recommend exploring cross-cultural variation in “feedback acceptance” and integrating engagement-type mappings, which are unrelated to my study’s objectives, data, and design. My work focuses specifically on measurable vocabulary and writing gains and on teachers’ and students’ perceptions within Pakistani ESL classrooms.

Given these discrepancies, the review report appears to be intended for another manuscript. Furthermore, the language, structure, and generic breadth of the comments strongly suggest that the report may have been generated with the assistance of AI tools. If that is the case, I respectfully request that the editorial office sensitise reviewers to avoid submitting uncritical AI-generated reports, as such feedback risks misrepresenting the work under review and undermines the peer-review process. For completeness, I would like to highlight that all the comments from Reviewer 4 were meticulously incorporated into the revised manuscript, ensuring that the feedback was addressed comprehensively and constructively.

Thank you very much for your attention to this matter and for your continued efforts in maintaining rigorous scholarly standards.

Sincerely,

Tahir Saleem

Professor, Department of English

University of Central Punjab, Lahore, Pakistan

Email: tahir.saleem@ucp.edu.pk

---

## [Decision Letter · Decision Letter 3]

15 Sep 2025

Integrating AI in Pakistani ESL Classrooms: Teachers' Practices, Perspectives, and Impact on Student Performance

PONE-D-25-13511R3

Dear Dr. Saleem,

We’re pleased to inform you that your manuscript has been judged scientifically suitable for publication and will be formally accepted for publication once it meets all outstanding technical requirements.

Kind regards,

Mostafa Janebi Enayat, Ph.D.

Academic Editor

PLOS ONE

Additional Editor Comments (optional):

Reviewer #5:

Reviewers' comments:

Reviewer's Responses to Questions

**Comments to the Author**

1. If the authors have adequately addressed your comments raised in a previous round of review and you feel that this manuscript is now acceptable for publication, you may indicate that here to bypass the “Comments to the Author” section, enter your conflict of interest statement in the “Confidential to Editor” section, and submit your "Accept" recommendation.

Reviewer #5: All comments have been addressed

2. Is the manuscript technically sound, and do the data support the conclusions?

Reviewer #5: Yes

3. Has the statistical analysis been performed appropriately and rigorously? 

Reviewer #5: Yes

4. Have the authors made all data underlying the findings in their manuscript fully available?

Reviewer #5: Yes

5. Is the manuscript presented in an intelligible fashion and written in standard English?

Reviewer #5: Yes

6. Review Comments to the Author

Reviewer #5: Dear authors

Thank you very much for addressing the issues raised by the reviewers.

The main issues addressed by reviewers were taken into account.

Cordially,

7. PLOS authors have the option to publish the peer review history of their article (what does this mean? ). If published, this will include your full peer review and any attached files.

**Do you want your identity to be public for this peer review?** For information about this choice, including consent withdrawal, please see our Privacy Policy .

Reviewer #5: No

---

## [Editor Report · Acceptance letter]

PONE-D-25-13511R3

PLOS ONE

Dear Dr. Saleem,

I'm pleased to inform you that your manuscript has been deemed suitable for publication in PLOS ONE. Congratulations! Your manuscript is now being handed over to our production team.

Kind regards,

on behalf of

Dr. Mostafa Janebi Enayat

Academic Editor

PLOS ONE